# Numerical Simulation of Single-Droplet Dynamics, Vaporization, and Heat Transfer from Impingement onto Static and Vibrating Surfaces

**J. Thalackottore Jose and J. F. Dunne ***

Department of Engineering and Design, School of Engineering and Informatics, University of Sussex, Falmer, Brighton BN1 9QT, UK; J.Thalackottore-Jose@sussex.ac.uk
* Correspondence: j.f.dunne@sussex.ac.uk

**Abstract:** A numerical study is presented to examine the behavior of a single liquid droplet initially passing through air or steam, followed by impingement onto a static or vibrating surface. The fluid dynamic equations are solved using the Volume of Fluid method, which includes both viscous and surface tension effects, and the possibility of droplet evaporation when the impact surface is hot. Initially, dynamic behavior is examined for isothermal impingement of a droplet moving through air, first without and then with boundary vibration. Isothermal simulations are used to establish how droplet rebound conditions and the time interval between initial contact to detachment vary with droplet diameter for droplet impingement onto a stationary boundary. Heat transfer is then assessed for a liquid droplet initially at saturation temperature passing through steam, followed by contact with a hot vibrating boundary, in which droplet evaporation commences. The paper shows that, for droplet impingement onto a static boundary, the minimum impact velocity for rebound reduces linearly with droplet diameter, whereas the time interval between initial contact and detachment appears to increase linearly with droplet diameter. With the introduction of a vibrating surface, the minimum relative impact velocity for isothermal rebound is found to be higher than the minimum impact velocity for static boundary droplet rebound. For impingement onto a hot surface, in which droplet evaporation commences, it is shown that large-amplitude surface vibration reduces heat transfer, whereas low-amplitude high-frequency vibration appears to increase heat transfer.

**Keywords:** liquid droplet; impingement; boundary vibration; bouncing; evaporation; heat transfer

---

## 1. Introduction

Single-phase liquid spray cooling is attractive for its high heat removal capabilities. Spray evaporative cooling is potentially even more attractive owing to its very high heat removal potential. Very high heat flux removal is essential for cooling of components that are internally generating high levels of heat. Pais et al. [1] measured, in laboratory conditions, spray boiling heat flux levels up to 12 MW/m$^2$. Evaporative cooling has already been used in several application areas, such as air coolers, power plant, and in some power electronics [2]. Spray evaporative cooling is, therefore, a potentially very promising candidate for cooling in several high-heat-generating application areas, including in very-high-current power electronics and highly downsized internal combustion engines. In both of these latter cases, existing cooling methods are struggling to cope with the very high levels of heat removal needed.

In a number of application areas, particularly associated with transport systems, the components that need to be cooled also vibrate significantly during operation. The effect of surface vibration on spray evaporative cooling is an area that has received very limited previous attention. Sarmadian et al. (2020) [3]

undertook an experimental investigation specifically on the surface vibration of spray evaporative cooling in the nucleate boiling regime. There have also been a number of experimental studies on evaporative cooling but not involving a spray. For example, Kim et al. (2002) [4] examined the effects of mechanical vibration on flow boiling; Atashi et al. (2014) [5] studied the effect of vibrations on pool boiling, whereas Sathyabhama and Prashanth (2015) [6] studied pool-boiling heat transfer enhancement using surface vibration. All of these studies examined particular effects within a specific set of parameters, with experimental verification limited to within those parameter ranges. Such experimental investigations require considerable resources and may not even be able to generate certain types of information.

In principle, numerical simulation does not have the same limitations as experimental investigation, and, if numerical tools can be exploited, they offer very important insight into the physics associated with vibration and its effect on spray boiling heat transfer. However, to simulate full spray evaporation, numerical simulation is massively expensive from a computational viewpoint. Numerical simulation of single-droplet impingement and evaporation, by contrast, is a much more practical option owing to its reduced complexity, the results of which can be compared with measured spray boiling heat transfer to check whether there is any correlation between the two. Wang et al. (2016) [7] simulated a droplet impinging onto a liquid film (on a moving boundary). They used a two-dimensional (2D) Volume of Fluid (VOF) method to simulate the effect of surface vibration at frequencies in the range of 200 Hz to 800 Hz and amplitudes in the range of 0.025 mm to 1 mm. The study showed a heat transfer increase of 8.1% at a frequency of 200 Hz for all amplitudes but a reduction in heat transfer of 58.2% at a frequency of 800 Hz.

To accurately simulate the fluid dynamics of droplet impingement, the contact angle that the droplet makes with the surface, at the three-phase contact line, must be accurately modeled [8–10]. When the droplet is stationary, it makes a stationary contact angle with the surface. However, when the droplet spreads across a surface, it has a dynamic contact angle. There are several theoretical models to predict the dynamic contact angle but all require the determination of contact line velocity. Several researchers conducted numerical studies with different dynamic contact angle models embedded in their numerical models. Sikalo et al. (2005) [8] reviewed several contact angle models for the dynamics of spreading droplets. A popular dynamic contact model is the empirical correlation by Kistler [8], who used an inverse Hoffman function based on capillary number. Another successful dynamic contact angle model is from Blake and Coninck (2002) [11]. Their simpler model combined the molecular kinetic theory of wetting, the out-of-balance surface-tension force, and the Frenkel–Eyring activated-rate theory of transport in liquids [12]. Both models [8,11] require the contact line velocity to be determined. One model complication generally stems from assuming a "no slip" condition at the liquid–solid interface. This creates a stress singularity at the interface during liquid movement on the surface. Cox (1985) [13] successfully developed a slip condition at the interface to remove the stress singularity.

The VOF method is a widely used fluid dynamic approach to model two or more fluids where there is no mixing. Chen et al. (2016) [14] numerically simulated droplet impingement dynamics and evaporation using the VOF method deploying the Blake and Coninck contact-angle calculation method [11]. They validated their predictions by comparison with experimentally measured droplet impingement data obtained by Dong et al. (2006) [15], showing good agreement. Briones and Ervin (2010) [16] compared VOF simulations with experimental measurements for both micrometer-sized droplet impingement dynamics and evaporation. Different contact-angle models were used, and it was found that Blake's [11] contact-angle model gave the best agreement with measurement. Sikalo et al. (2005) [8] used VOF numerical simulations of droplet impingement dynamics, where a 2D axisymmetric model was used, with three-dimensional (3D) simulations used to check its accuracy. The 2D axisymmetric model simulations compared well with the 3D model. Chen et al. (2016) [14] also successfully used a 2D axisymmetric model to simulate droplet impingement and found that significant time savings can be achieved in comparison to a 3D problem.

In this paper, the dynamics and evaporation of a droplet impinging onto a vibrating boundary are simulated, using the VOF method. A 2D axisymmetric model as utilized to ensure computational

efficiency using the Blake and Coninck contact-angle model [11] with respect to the contact line velocity. A new approach is developed and implemented in the model to determine the contact line velocity when a non-slip condition is applied. The proposed simulation model is first validated against published experimental data and then to understand the mechanisms involved in (i) droplet rebound, (ii) impingement, and (iii) evaporation on fixed and moving boundaries. The objectives were to assess the computational feasibility of simulating droplet impingement in terms of accuracy and efficiency.

## 2. Numerical Model

Here, the governing equations are stated, which are needed to simulate a water droplet making contact with a hot vibrating boundary under the influence of gravity. The droplet is initially liquid moving through either air or steam but then vaporizes when contact is made with the hot vibrating boundary. The model comprises continuity, momentum, and energy equations for a viscous fluid with surface tension, but also includes an appropriate phase change model. These continuum equations are solved using the Volume of Fluid method (via ANSYS-FLUENT version 19.2) within a rectangular domain. The wall-contact dynamic model and the specified boundary conditions are explained below, followed by domain and mesh size details plus solution control information. The model was initially verified by comparison with published experimental measurements for droplet impingement onto a static boundary.

### 2.1. Volume of Fluid Method

The Volume of Fluid Method models the interaction of immiscible liquids by solving the continuity equation associated with a volume fraction to enable phase tracking [17]. The continuity equation takes the following form:

$$\frac{1}{\rho_q}\left[\frac{\partial}{\partial t}\left(\alpha_q\rho_q\right) + \nabla\cdot\left(\alpha_q\rho_q\vec{v}_q\right) = S_{\alpha_q} + \sum_{p=1}^{n}\left(\dot{m}_{pq} - \dot{m}_{qp}\right)\right], \tag{1}$$

where $\rho$ is the density, $\alpha$ is the volume fraction, $\vec{v}$ is the velocity vector, $\dot{m}_{pq}$ is the mass transfer from phase $p$ to phase $q$, $\dot{m}_{qp}$ is the mass transfer rate from phase $q$ to $p$, and $S_{\alpha_q}$ is a mass source term resulting from evaporation where suffix $q$ is an identifier for the fluid. Tracking of the interface was made possible by assigning a volume fraction to each fluid and adding the volume fractions to give unity. The gaseous phase, which is steam, was assigned as the primary phase, which in turn made water the secondary phase. All the properties and variables of each phase were calculated by volume averaging. The volume fraction of the secondary phase was not directly solved, but was calculated from the volume fraction obtained for the primary phase. The volume fraction was obtained (in ANSYS) via an explicit time solution owing to its known higher accuracy compared to an implicit formulation.

The momentum equation,

$$\frac{\partial}{\partial t}\left(\rho\vec{v}\right) + \nabla\cdot\left(\rho\vec{v}\vec{v}\right) = -\nabla P + \nabla\cdot\left[\mu(\nabla\vec{v} + \nabla\vec{v}^{T})\right] + \rho\vec{g} + \vec{F}, \tag{2}$$

where $P$ is the pressure, $\mu$ is the dynamic viscosity, and $\vec{F}$ is the body force vector, was solved throughout the domain and between the phases, for which the velocity field was shared. By enabling the Continuum Surface Force (CSF) model (in ANSYS, which is based on the model by Brackbill et al. [18]), this provided a surface-tension component for body force calculation at the liquid–gas interface. To determine the surface tension from the pressure difference across the interface and the surface curvature, the solution was obtained from

$$p_2 - p_1 = \sigma\left(\frac{1}{R_1} + \frac{1}{R_2}\right), \tag{3}$$

where $p_1$ and $p_2$ are the pressures on either side of the surface, $\sigma$ is the surface tension, and $R_1$ and $R_2$ are the two radii perpendicular to each other used for measuring the surface curvature. The surface curvature ($k$) was computed in terms of the divergence of the unit normal ($\hat{n}$) at the interface as follows:

$$k = \nabla \cdot \hat{n}. \tag{4}$$

The unit normal was calculated as

$$\hat{n} = \frac{n}{|n|}, \tag{5}$$

where $n$ is the surface normal computed as the gradient of the liquid-phase volume fraction ($\alpha_l$),

$$n = \nabla \cdot \alpha_l. \tag{6}$$

A source term was added to the momentum equation by switching on the CSF model defined as

$$F_\sigma = \sigma_{lg} \frac{\rho k_l \nabla \alpha_l}{\frac{1}{2}(\rho_l + \rho_g)}, \tag{7}$$

where $k_l$ is the surface curvature, $\rho$ is the volume-averaged density, and the subscripts $l$ and $g$ denote liquid and gas phases, respectively.

To specify a time-dependent contact angle at the three-phase contact line, the default wall adhesion model was used, based on Brackbill et al. [18]. The surface normal vector $\hat{n}$, adjacent to the wall, was modified by the wall adhesion model, providing the surface curvature of the interface at the contact line. The surface normal was given by

$$\hat{n} = \hat{n}_w \cos \theta_w + \hat{t}_w \sin \theta_w, \tag{8}$$

where $\theta_w$ is the contact angle, and $\hat{n}_w$ and $\hat{t}_w$ are the unit vectors normal and tangential to the wall [17]. The energy equation determined the heat transfer, which was defined as:

$$\frac{\partial}{\partial t}(\rho E) + \nabla \cdot \left( \vec{v}(\rho E + p) \right) = \nabla \cdot (k\nabla T) + S, \tag{9}$$

where $E$ is the energy, $T$ is the temperature, $k$ is the thermal conductivity, and $S$ is the source term for phase change contributions from the interface [14]. The energy $E$ and the temperature $T$ were used as mass-averaged variables. The mass-averaged energy, for example, was obtained from

$$E = \frac{\sum_{q=1}^{n} \alpha_q \rho_q E_q}{\sum_{q=1}^{n} \alpha_q \rho_q}, \tag{10}$$

where density, $\rho$, and thermal conductivity, $k$, are volumetrically averaged properties.

### 2.2. The Phase Change Model

The Lee model [19] for evaporation–condensation was employed for phase change mass transfer, where the mass transport equation is defined as

$$\frac{\partial}{\partial t}(\alpha_v \rho_v) + \nabla \cdot \left( \alpha_v \rho_v \vec{V}_v \right) = \dot{m}_{lv} - \dot{m}_{vl}, \tag{11}$$

where $\alpha_v$ is the vapor volume fraction, $\rho_v$ is the vapor density, $\vec{V}_v$ is the vapor phase velocity, $\dot{m}_{lv}$ is the rate of mass transfer from liquid to vapor, and $\dot{m}_{vl}$ is the rate of mass transfer from vapor to liquid. The respective evaporation and condensation mass transfer terms in Equation (8) are

$$\dot{m}_{lv} = C_e \alpha_l \rho_l \frac{T_l - T_{sat}}{T_{sat}},\tag{12}$$

and

$$\dot{m}_{vl} = C_c \alpha_v \rho_v \frac{T_{sat} - T_v}{T_{sat}},\tag{13}$$

where $C_e$ and $C_c$ are coefficients to represent evaporation and condensation, and where subscripts $l$, $v$, and *sat* stand for liquid, volume, and saturation, respectively (a default value of 0.1 was used for both coefficients). The coefficients required to accurately predict the evaporative mass transfer may vary. In order to determine appropriate coefficients, simulations need to be undertaken and verified by comparison with experimentally measured data. Chen et al. [2] used a mass transfer model based on kinetic theory, where it was stated that the evaporation coefficient was not previously accurately measured. They used a value of 0.1 in [2] but stated that it could be in the range 0 to 1. Briones and Ervin [3] undertook both experimental measurements and numerical simulations on micrometer-sized droplet impingement and evaporation. They used a mass transfer model based on interfacial surface area density and kinetic theory. It has generally been reported that there are considerable differences between theoretical and experimentally determined mass transfer coefficients. One of the drawbacks in verifying numerical simulations using experimental data is the enormous computational power needed to simulate complete droplet evaporation. It was reported in [3] that their single-droplet evaporation simulation took 24 days of computer time using parallel processing.

In the present study, it was not feasible with the available computing power to simulate complete droplet evaporation with varying coefficients in order to determine the most accurate coefficient needed to produce results closest to experimental measurements. For example, simulating droplet impingement and evaporation for a physical time duration of 1 ms takes 4 days on a personal computer (PC) with an Intel i7 processor. Tens of milliseconds of physical time are needed to achieve full droplet evaporation, as reported by Chen et al. [2] for a droplet size of 49 μm. A simulation involving complete droplet evaporation was, therefore, estimated to take several months on an Intel i7 processor. For this reason, the default coefficient values were used, and a study to determine the most appropriate coefficients was not attempted

## 2.3. Numerical Domain

A 2D axisymmetric model within a 2D square geometry (instead of a 3D model) was used to avoid excessive computational time, since a 2D model is known to give a wholly accurate approximation. A mesh with quadrilateral elements within the 2D square was chosen to represent the domain enclosed by walls in which the vertical left-hand-side wall represents the impact boundary. The droplet was modeled as a semicircle, with a defined axis of symmetry as shown in Figure 1.

When the geometry is reflected around the horizontal axis, it forms a hollow cylinder with a droplet at a specified distance from the impact surface. To approximate temperature-dependent liquid and gas properties across the domain, a piecewise-linear fit to tabulated property data was deployed. After flow-field initialization, where initial values of pressure, temperature, velocity, and volume fraction were provided, the volume fraction and velocity were "patched on" to a circular cell region marked for the size of droplet. "Patching" is a term used to describe the assignment of an area in a domain. In the case of a droplet, this was a semi-circle. Droplets were actually "patched" 2 μm above the impact surface to provide space for the surrounding fluid to move to, i.e., to ensure non-zero velocity resulting from droplet travel before impact.

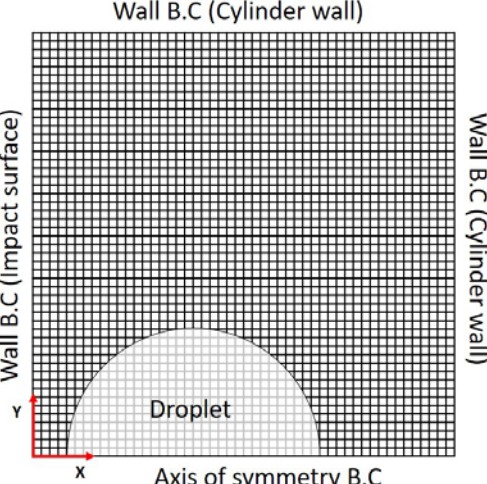

**Figure 1.** Numerical domain of the axisymmetric model for a droplet traveling from right to left (showing a very unrefined mesh).

The chosen explicit volume fraction formulation used a different sub-time-step from the other transport equations. This different sub-time-step was calculated from a separate Courant number. The so-called hybrid method, which uses a combination of velocity and flux average, was chosen to determine the volume fraction Courant number specification [19]. The hybrid method determines a sub-time-step small enough to capture both velocity and flux, especially near to the wall on which a droplet impacts, thereby providing better convergence.

For spatial discretization, the second-order upwind scheme was used owing to its higher accuracy [20]. The "face fluxes" were interpolated using the geometric reconstruction scheme (i.e., via the Geo-Reconstruct option in ANSYS FLUENT), namely, when a cell is near to the interface between two faces, where use of a piecewise-linear method assumes a linear interface slope within a cell [17]. (The "face flux" describes the flux through a face of an element or a volume). Geo-Reconstruct is the most accurate interpolation method in ANSYS FLUENT for multiphase problems [17].

The pressure-based solver along with the SIMPLEC algorithm was used for pressure–velocity coupling. The least squares cell-based method was chosen for gradient calculations [20] to minimize computation time and to provide sufficient accuracy compared with more superior computationally expensive methods. To interpolate the control volume face pressure, the body force weighted scheme was chosen owing to its superior performance with VOF simulations involving fluids of large differences in density [14]. The first-order implicit transient formulation was enabled for the temporal discretization of the governing equations [20].

*2.4. Wall-Contact Dynamics*

The Blake and Coninck dynamic contact-angle model [11] was implemented at the wall boundary using appropriate user defined functions (UDFs). The model [11] requires the contact line velocity to determine the dynamic contact angle, which has the following form:

$$v_{cl} = \frac{2k_s h \lambda}{\mu_l v_l} \sinh\left[\frac{\sigma_l}{2n k_b T}\left(\cos \theta^0 - \cos \theta^d\right)\right], \tag{14}$$

where $v_{cl}$ is the contact line velocity, $k_s$ is the frequency of molecular displacements, $h$ is Plank's constant, $\lambda$ is the length of each molecular displacement, $\mu_l$ is the dynamic viscosity of the liquid, $v_l$ is the molecular volume, $n$ is the number of absorption sites per unit length, $k_b$ is the Boltzmann's

constant, $T$ is the temperature of the liquid, $\theta^0$ is the static contact angle, and $\theta^d$ is the dynamic contact angle. Rearranging Equation (14) for a dynamic contact angle gives

$$\theta^d = \cos^{-1}\left(\cos\theta^0 - \left[\frac{2nk_bT}{\sigma_l}\sinh^{-1}\left(v_{cl}\frac{\mu_l v_l}{2k_s h\lambda}\right)\right]\right). \tag{15}$$

Numerical values of the properties used in Equation (15), for water as the wetting liquid (as given in Chen et al. (2016) [14]), are given in Table 1.

**Table 1.** Parameter values used in the dynamic contact angle calculation (Equation (11)).

| Property | Value |
|---|---|
| Frequency of molecular displacements, $k_s$ | $4.5276 \times 10^{10}$ Hz |
| Plank's constant, $h$ | $6.62607004 \times 10^{-34}$ kg/s |
| Length of each molecular displacement, $\lambda$ | $0.5 \times 10^{-9}$ m |
| Molecular volume, $v_l$ | $3 \times 10^{-29}$ m³ |
| Number of absorption sites per unit length, $n$ | $4 \times 10^{18}$ m$^{-1}$ |
| Boltzmann's constant, $k_b$ | $1.38064852 \times 10^{-23}$ J·K$^{-1}$ |

The maximum wetting velocity and the minimum de-wetting velocity are given as

$$v_{cl180} = \frac{2k_s h\lambda}{\mu_l v_l}\sinh\left[\frac{\sigma_{lv}}{2nk_bT}\left(\cos\theta^0 + 1\right)\right], \tag{16}$$

and

$$-v_{cl0} = \frac{2k_s h\lambda}{\mu_l v_l}\sinh\left[\frac{\sigma_{lv}}{2nk_bT}\left(1 - \cos\theta^0\right)\right]. \tag{17}$$

To define a dynamic contact angle B.C. at the droplet impingement wall, two different UDFs were used. One UDF was needed to calculate the velocity of the contact line; the other UDF was to input the contact angle B.C. The contact line velocity $v_{cl}$, calculated by one of the UDFs, was used by the other UDF to determine the dynamic contact angle $\theta^d$, using Equation (15). Equations (16) and (17) were used to determine the maximum wetting velocity and the minimum de-wetting velocity, respectively. For velocities above the maximum allowable wetting velocity, a contact angle of 180° was applied, and, for velocities below the minimum allowable de-wetting velocity, a contact angle of 0° was applied.

The UDF codes were written in C code using macros supplied by ANSYS called the DEFINE macros. The DEFINE macros enable real-time output and input access to simulation domain properties whilst the simulation is running. Two different types of DEFINE macro were used, DEFINE_ADJUST and DEFINE_PROFILE. The DEFINE_ADJUST macro allows modification of ANSYS Fluent variables that are not directly passed as arguments. This means that the variable and the domain from which it needs access should be specified using identifiers called "threads". The DEFINE_PROFILE macro enables implementation of custom boundary profiles which vary with time and space [21].

Using the DEFINE_ADJUST macro, a code was written to calculate the contact line velocity. In ANSYS FLUENT, a "no slip" condition was applied with the wall adhesion model. This sets the velocity components of the fluid in contact with the wall to zero. However, the three-phase contact line moves during the spreading and de-spreading of the droplet. A new method was developed to solve this problem. The droplet interface surface was assumed to be like a sheet that unrolls and rolls over a surface. This essentially satisfies the rule of zero velocity of fluid in contact with the wall, allowing movement of the contact line. The surface of the sheet in contact with the wall has a velocity of zero but when the sheet unrolls or rolls over the surface, the boundary line (contact line) of the area in contact with the surface moves. A cell zone was created in the domain for the row of cells adjacent to the cells attached to the wall. The $y$-direction velocity components in the cell zone created were accessed from cells with a density higher than 500 kg/m³. This ensured that only the velocity of the

liquid was retrieved. The ANSYS predefined C_V(c,t) and C_R(c,t) macros were employed to access the *y*-velocity and the density in the cells, respectively [21]. The velocity with the highest magnitude was chosen as the contact line velocity from the velocities accessed. From the contact line velocity calculated, another code using the DEFINE_PROFILE macro calculated the dynamic contact angle using Equation (15) and applied it at the wall as a B.C. at the beginning of each time-step. The code, based on the maximum wetting velocity and the minimum de-wetting velocity, also sets the maximum and minimum contact angles, respectively. A third UDF was used to apply the vibration model to the wall, which used the DEFINE_PROFILE macro to apply a velocity component to the wall with respect to the simulation physical time. The wall vibration movement had a sinusoidal wave form as a function of the amplitude and frequency specified. Figure 2 shows a schematic representation of the velocity output for contact line velocity calculation, dynamic contact application, and wall vibration.

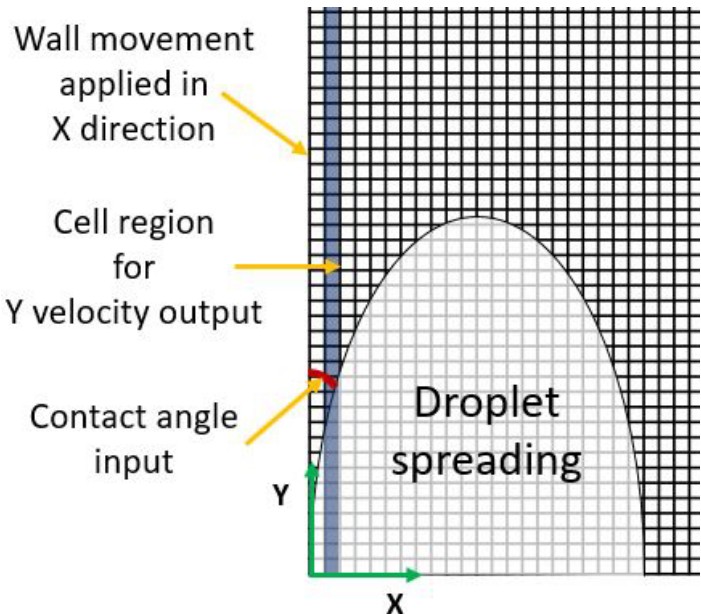

**Figure 2.** Schematic showing the user-defined function (UDF) inputs and outputs.

### 2.5. Domain Mesh-Size Selection and Solution Control

Four different domain sizes were modeled for four different droplet sizes. Each domain was just large enough to contain the droplet during the spreading and rebound phase to minimize the computational time. For droplet diameters of 30 μm, 49 μm, 75 μm, and 100 μm, the corresponding side lengths of the square domain were 60 μm, 100 μm, 150 μm, and 200 μm, respectively. The domain interior used an element mesh size of 0.5 μm with boundary conditions applied. A global Courant number of 0.5, with variable time-stepping, was employed for the time-step calculation, whereas, for determining the sub-time-step for the volume fraction equation, a Courant number of 0.25 was used. Variable (very small) time-stepping was used to efficiently provide sufficient accuracy to capture the highly transient nature of droplet impingement dynamics. An initial time-step of 10 ns was specified, along with minimum and maximum time-steps of 1 ns and 1 μs, and the maximum number of iterations per time-step was set to 20.

### 2.6. Validation of Model

Dong et al. [15] recorded (using a high-speed camera) experimental inkjet drop formation and deposition, successfully capturing in great detail impingement dynamics of a 49 μm diameter droplet impacting on gold-coated silicon wafers at a velocity of 4 m/s, and having a liquid–wall static contact angle of 110°. To undertake a numerical simulation of the impingement conditions in [15], the developed model included a water droplet of the same diameter, impact velocity, and static contact

angle. The ambient temperature needed to define some of the simulation fluid properties was assumed to be 20 °C. Two meshes with different global element sizes of 1 µm and 0.5 µm were used to validate the simulations. Figure 3 shows as a function of time the simulated evolution of the dimensionless spreading diameter $D$ compared with the diameter experimentally measured in [15].

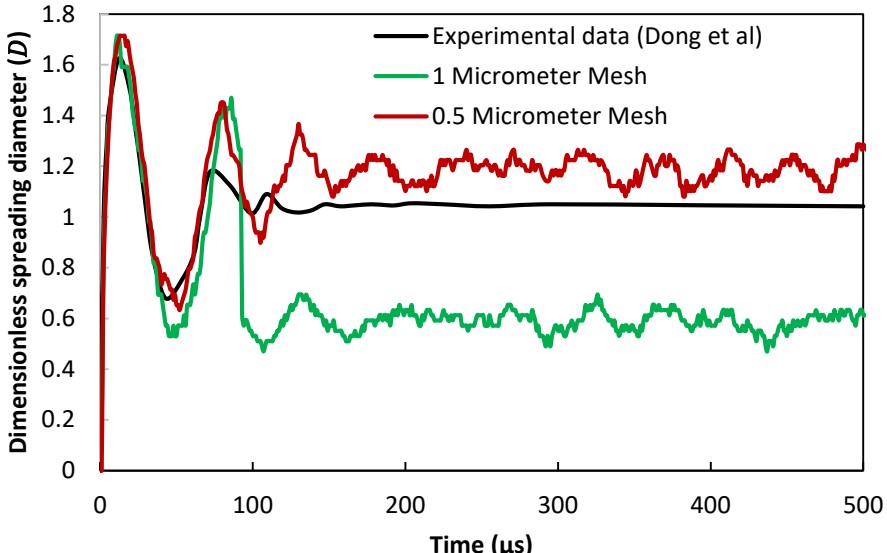

**Figure 3.** Dimensionless spreading diameter: simulation compared with experimental measurement [15].

Figure 3 shows that the dimensionless spreading diameter using a 0.5 µm mesh is generally in good agreement with the measurement even though the simulated diameter does not appear to reach a steady state. This difference may actually stem from measurement difficulties in capturing minute movements of the droplet spreading diameter, whereas the simulation captures movements down to a length of 0.5 µm. The 0.5 µm mesh simulations took approximately 18 h to simulate 500 µs of physical duration on an Intel i7 processor workstation with eight cores and 16 GB random-access memory (RAM). The mesh size could obviously be reduced for improved accuracy, but the simulation running time would increase disproportionately. Figure 4 shows, using a mesh size of 0.5 µm, the temporal evolution of the droplet shape obtained from simulation compared with experimental measurement [15].

Figure 4 shows that the simulated droplet shape is very similar to the experimentally measured shape with the only exception being a microsecond delay in simulated droplet dynamics. This delay is evident from a comparison of the droplet shapes at 10 µs and 20 µs, where the experimentally measured droplet shows flattening and oscillating back in shape at least 1 µs earlier. Validation of droplet evaporation was not attempted owing to computational demand because it takes tens of milliseconds for complete droplet evaporation [14]. The simulations were restricted to a physical time of 1000 µs where the onset of evaporation happens. This timescale was established from running simulations with and without wall vibration, which took between 2 and 4 days to simulate a physical duration of 1000 µs, depending on the vibration amplitude and frequency. Figure 5 shows, from simulating a 49 µm diameter droplet, that evaporation commenced after a physical time of 500 µs and continued until 1000 µs after impinging onto a hot surface, at a velocity of 1 m/s. The initial temperature of the droplet was 100 °C, whereas the temperature of the wall was constant at 150 °C throughout.

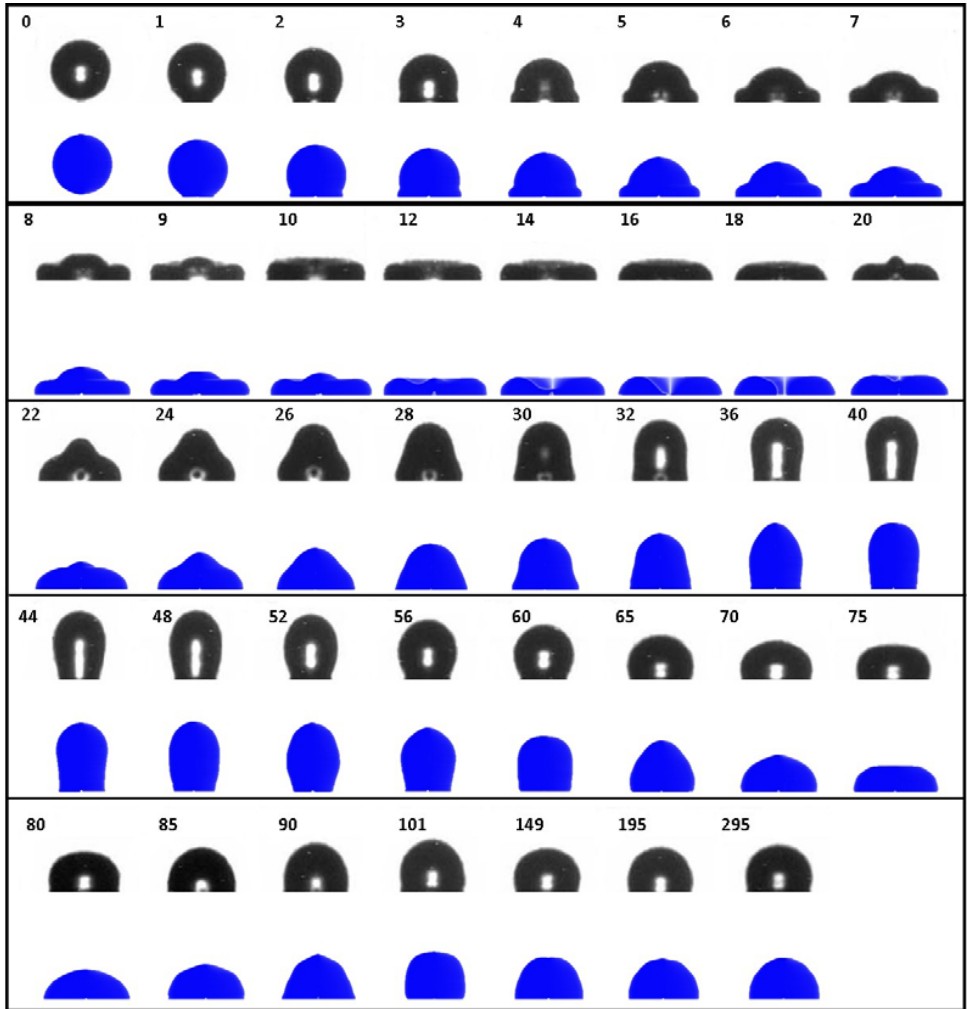

**Figure 4.** Temporal evolution of droplet shape during impingement at various discrete times up to 295 μs: experimentally measured (black) [15]; numerical simulation (blue).

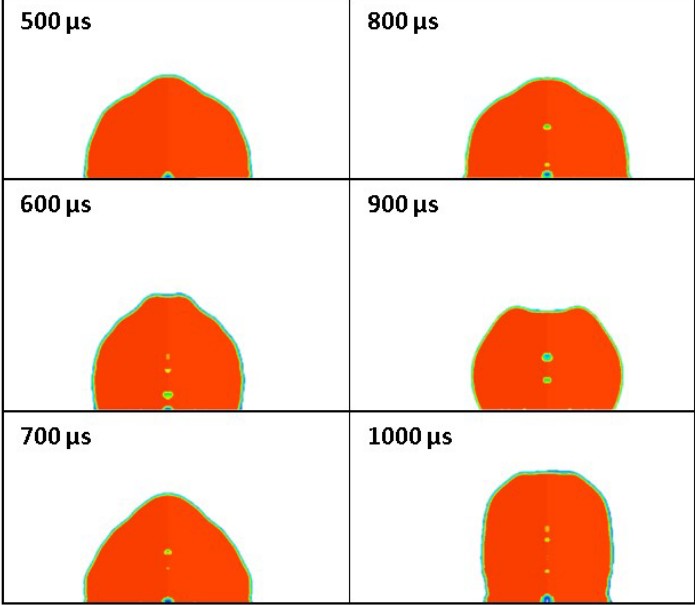

**Figure 5.** Simulated droplet evaporation shown commencing after impact.

## 3. Simulation of Droplet Impingement onto a Hot Vibrating Boundary

Getting an understanding of droplet rebound when impinging onto a hot vibrating boundary is of practical importance for spray evaporative cooling. In particular, knowledge of the droplet velocities and vibration conditions that cause rebound should give insight into the conditions where a reduction in heat transfer will occur. Simulations using the model described in Section 3 were, therefore, considered in four different scenarios: (i) isothermal droplet impingement onto a stationary boundary, (ii) isothermal impingement onto a moving boundary, (iii) impingement, with the onset of evaporation onto a stationary boundary, and (iv) impingement, with the onset of evaporation onto a moving boundary. The set of relevant simulations parameters corresponding to scenarios (i) to (iv) are shown in Table 2.

**Table 2.** Droplet parameters and boundary frequencies and amplitudes used in the simulations. N/A, not applicable.

| Simulation No. | D (μm) | V (m/s) | A (mm) | F (Hz) |
|:---:|:---:|:---:|:---:|:---:|
| **Isothermal Impingement onto a Stationary Boundary** | | | | |
| A1 | 30 | 6 | N/A | N/A |
| A2 | 30 | 7 | N/A | N/A |
| A3 | 30 | 7.5 | N/A | N/A |
| A4 | 49 | 5 | N/A | N/A |
| A5 | 49 | 6 | N/A | N/A |
| A6 | 49 | 6.5 | N/A | N/A |
| A7 | 75 | 3.5 | N/A | N/A |
| A8 | 75 | 4.5 | N/A | N/A |
| A9 | 100 | 3 | N/A | N/A |
| A10 | 100 | 3.5 | N/A | N/A |
| **Isothermal Impingement onto a Moving Boundary** | | | | |
| B1 | 49 | 6.5 | 10 | 10 |
| B2 | 49 | 6.5 | 10 | 100 |
| B3 | 49 | 6.5 | 10 | 1000 |
| B4 | 49 | 6.5 | 1 | 10 |
| B5 | 49 | 6.5 | 1 | 100 |
| B6 | 49 | 6.5 | 1 | 1000 |
| B7 | 49 | 6 | 10 | 10 |
| B8 | 49 | 6 | 10 | 100 |
| B9 | 49 | 6 | 10 | 1000 |
| B10 | 49 | 6 | 1 | 10 |
| B11 | 49 | 6 | 1 | 100 |
| B12 | 49 | 6 | 1 | 1000 |
| **Impingement onto a Stationary Boundary with Onset of Evaporation** | | | | |
| C1 | 49 | 4 | 0 | 0 |
| C2 | 49 | 1 | 0 | 0 |
| **Impingement onto a Moving Boundary with Onset of Evaporation** | | | | |
| D1 | 49 | 1 | 10 | 10 |
| D2 | 49 | 1 | 10 | 100 |
| D3 | 49 | 1 | 10 | 1000 |
| D4 | 49 | 1 | 1 | 10 |
| D5 | 49 | 1 | 1 | 100 |
| D6 | 49 | 1 | 1 | 1000 |
| D7 | 49 | 1 | 0.1 | 10 |
| D8 | 49 | 1 | 0.1 | 100 |
| D9 | 49 | 1 | 0.1 | 1000 |

The frequencies and amplitudes used for moving boundary conditions in Table 2 actually correspond to the sort of agitation and vibration conditions experienced onboard automotive vehicles, particularly the dynamic conditions experienced by power electronics and batteries in electric vehicles, all of which must be thermally managed carefully and efficiently.

The isothermal stationary-boundary simulations were used to establish a criterion for droplet rebound and to determine the detachment time for different droplet sizes and impingement velocities. The isothermal moving-boundary simulations were designed to examine the effect of wall vibration on droplet rebound. Both sets of isothermal simulations used water in the presence of air as the droplet liquid. The properties of air and water were initially set to correspond to atmospheric pressure at a temperature of 20 °C, and a static contact angle of 110° was used for dynamic calculations.

The non-isothermal simulations, involving impingement, with onset of evaporation, were designed to understand heat transfer from droplet impingement and evaporation, first without and then with boundary vibration. The non-isothermal simulations used water as the droplet fluid and steam at as the surrounding gas corresponding to the vapor conditions in a closed chamber—both initially at 100 °C and atmospheric pressure. The (hot) wall was fixed at constant temperature of 150 °C. Under these conditions, almost all of the heat energy removed from the hot surface took place through droplet evaporation as latent heat. The wall was assumed to be aluminum. When in contact with the hot surface, the water droplet formed a static contact angle of 90° [22].

## 3.1. Isothermal Impingement onto a Hot Stationary Boundary

Simulation results for isothermal droplet rebound onto a stationary boundary are now given for droplet diameters of 30 μm, 49 μm, 75 μm, and 100 μm. In particular, the minimum impact velocities at which droplet rebound occurs were of interest. Figure 6 shows the droplet shapes leading to droplet detachment for the four different droplet diameters for microsecond time intervals between the shapes shown.

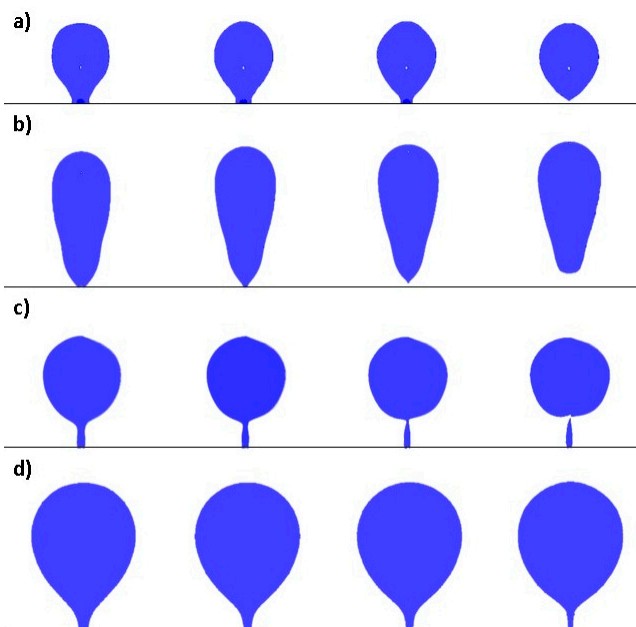

**Figure 6.** Changing droplet shape at time intervals of 30 μs, 44 μs, 115 μs, and 163 μs from initial boundary contact to rebound detachment for diameters of (**a**) 30 μm, (**b**) 49 μm, (**c**) 75 μm, and (**d**) 100 μm.

The simulated rebound velocities corresponding to the chosen diameters were 7.5 m/s, 6.5 m/s, 4.5 m/s, and 3.5 m/s, respectively. The corresponding physical time intervals from initial boundary contact to droplet rebound detachment were 30 μs, 44 μs, 115 μs, and 163 μs, respectively. The corresponding Weber numbers for each case were 23.16, 28.41, 20.84, and 16.81, respectively,

whereas the corresponding Reynolds numbers were 224.15, 317.30, 336.22, and 348.68. Breaking of the droplet or a turbulent movement was not observed in any of the cases, which shows that neither the critical Weber number nor the critical Reynolds number were reached. A point could be reached where both scenarios could happen with increasing droplet impingement velocity, but this tendency was not observed for the simulated impingement velocities.

Figure 7 shows the minimum droplet impact velocity for rebound to occur as a function of droplet diameter fitted (showing a linear least-squares fit through the discrete points).

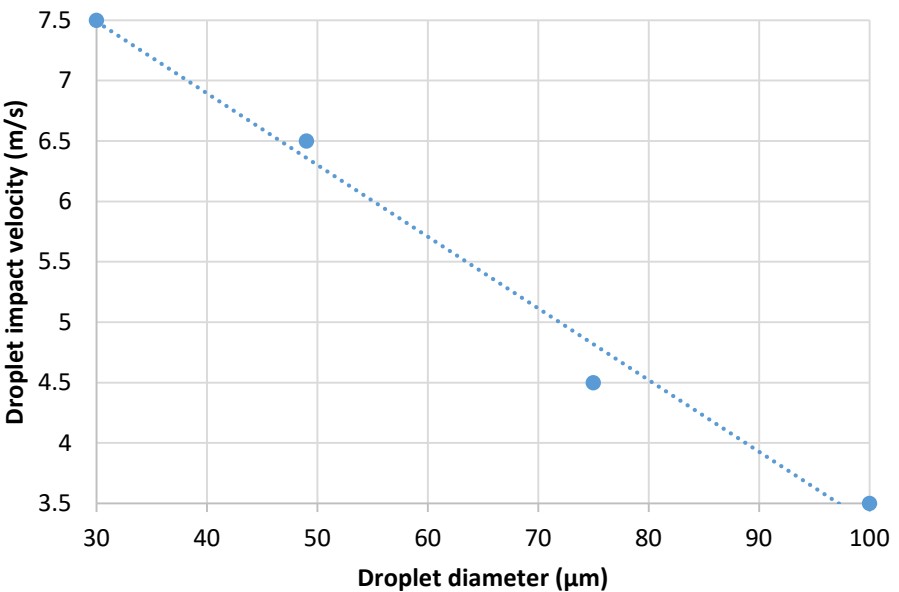

**Figure 7.** Minimum impact velocity to achieve droplet rebound as a function of droplet diameter.

Figure 7 shows that the minimum impact velocity for rebound reduced almost linearly with droplet diameter. Figure 8 shows the time interval from boundary contact to droplet detachment (after impingement) as a function of droplet diameter and a corresponding linear fit.

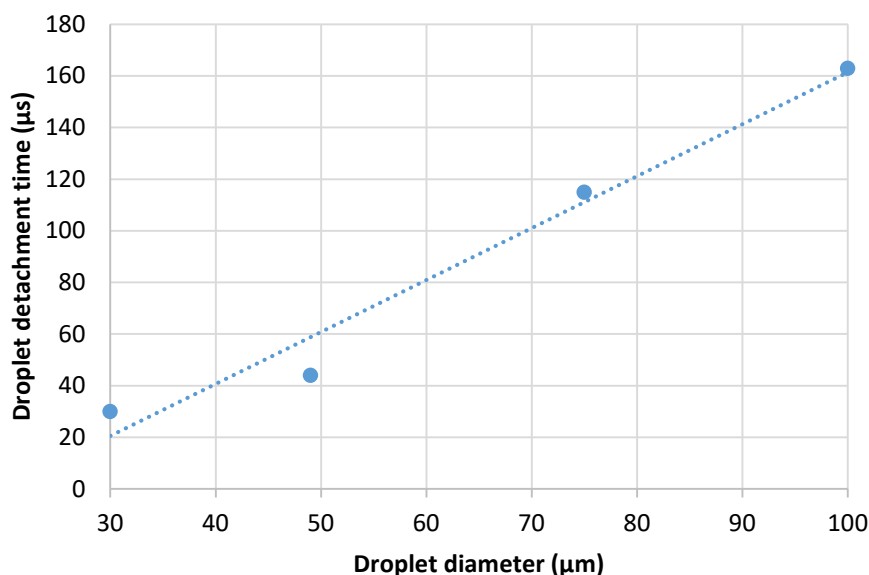

**Figure 8.** Time interval from boundary contact to droplet detachment as a function of droplet diameter.

Within the range of droplet diameters simulated, Figure 7 shows that the time interval from boundary contact to droplet detachment increased almost linearly with droplet diameter. For the

smallest droplet with a diameter of 30 μm, the droplet detachment time was 30 μs, whereas, for the largest droplet with a diameter of 100 μm, the time of detachment was 163 μs.

### 3.2. Isothermal Impingement onto Moving Boundary

Simulations to establish the effect of a moving boundary on droplet rebound at 0.5 m/s below the rebound minimum impact velocity ($v_{re}$) were of initial interest. A droplet of mid-range diameter 49 μm was chosen; the corresponding results are shown in Table 3.

**Table 3.** Simulation results for isothermal droplet impingement onto a moving boundary. Y, yes; N, no.

| Simulation No. | V (m/s) | A (mm) | F (Hz) | $V_{Wall}$ (m/s) | Rebound (Y/N) | $t_{re}$ (μs) |
|---|---|---|---|---|---|---|
| B1 | 6.5 | 10 | 10 | 0.628 | Y | 60 |
| B2 | 6.5 | 10 | 100 | 6.28 | Y | 55 |
| B3 | 6.5 | 10 | 1000 | 62.8 | Y | 57 |
| B4 | 6.5 | 1 | 10 | 0.0628 | N | N/A |
| B5 | 6.5 | 1 | 100 | 0.628 | Y | 60 |
| B6 | 6.5 | 1 | 1000 | 6.28 | Y | 60 |
| B7 | 6 | 10 | 10 | 0.628 | Y | 58 |
| B8 | 6 | 10 | 100 | 6.28 | Y | 56 |
| B9 | 6 | 10 | 1000 | 62.8 | Y | 42 |
| B10 | 6 | 1 | 10 | 0.0628 | Y | 60 |
| B11 | 6 | 1 | 100 | 0.628 | N | N/A |
| B12 | 6 | 1 | 1000 | 6.28 | N | N/A |

It is evident from the results in Table 3 that wall vibration influences droplet rebound. In fact, except for three vibration conditions, the simulated droplet rebounded at a velocity at and below the minimum velocity for droplet rebound onto a stationary boundary. For droplet simulation numbers B4 (1 mm, 10 Hz), B11 (1 mm, 100 Hz), and B12 (1 mm, 1000 Hz), rebound did not occur. In addition, for most of the cases, the time interval to droplet detachment was significantly increased in comparison to the stationary case of 44 μs, as shown in Figure 7. One possible reason for droplet rebound under moving boundary conditions occurring at a velocity below the stationary boundary minimum rebound velocity is that the oscillating boundary, on its upward movement at the beginning of its sinusoidal path, produced an impact velocity higher than the minimum velocity for rebound. However, this argument appears not to hold for the wall vibration amplitude of 1 mm, where, for simulation B10 (10 Hz), the droplet rebounded at a resultant velocity lower than the impact velocity for stationary rebound. The physical phenomenon associated with this behavior is not yet understood. Furthermore, in cases B11 (100 Hz) and B12 (1000 Hz), droplet rebound did not occur even though the resultant velocity was higher than required for rebound. These results show that rebound in the presence of a moving boundary is a complex phenomenon even under isothermal conditions.

### 3.3. Non-Isothermal Impingement and Onset of Evaporation onto a Fixed Boundary

Further simulations involving a droplet size of 49 μm were of interest but now involving phase change. Starting with droplet impingement onto a stationary boundary, the focus was now placed on the heat transfer from the hot surface into the droplet and then into the surrounding vapor. The conditions, such as viscosity and surface tension, were significantly different from the isothermal cases in so far as the liquid properties were necessarily set initially at 100 °C (rather than 20 °C of the isothermal simulations). The density, viscosity, surface tension, and thermal conductivity of the liquid were set to 958.35 kg/m$^3$, 0.28158 mPa·s, 0.0589 N/m, and 0.677 W/mK, respectively. The droplet velocity for non-isothermal simulations was reduced to 1 m/s because, in the initial studies, droplet rebound occurred when the droplet velocity was 4 m/s. Figure 9 shows the surface-averaged heat flux and a corresponding linear fit over the entire time duration. Figure 10 shows the dimensionless droplet spreading diameter with respect to time.

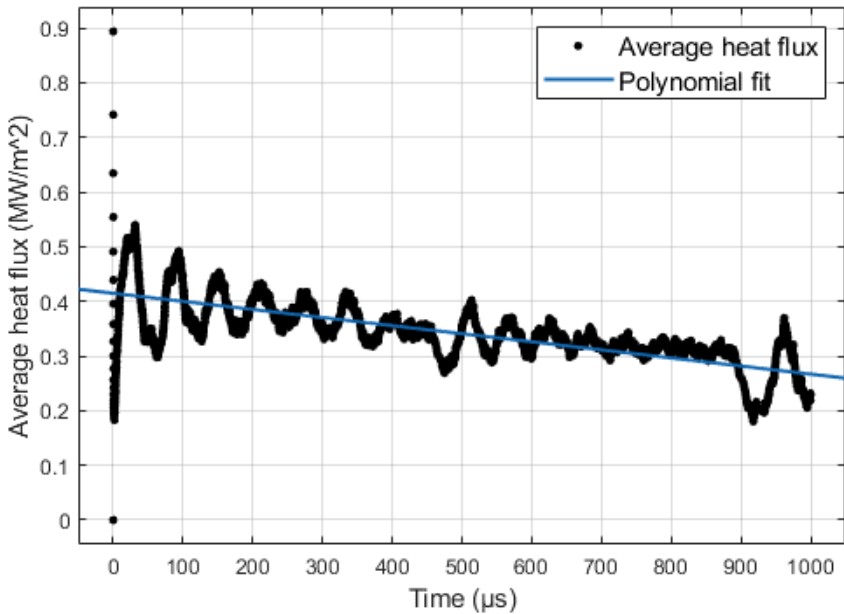

**Figure 9.** Surface-averaged heat flux via simulation of single-droplet impingement at a velocity of 1 m/s onto a stationary wall.

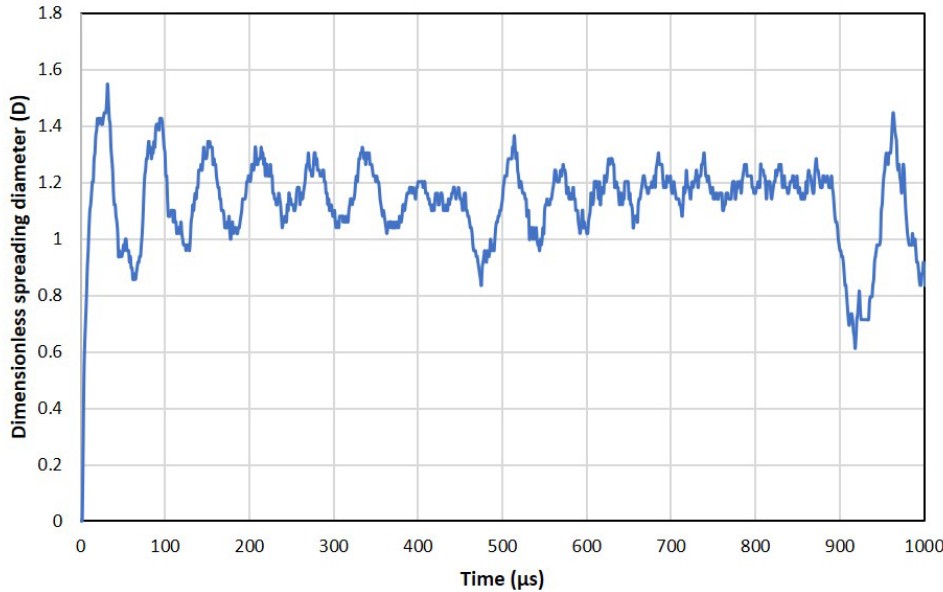

**Figure 10.** Droplet oscillation via simulation of single-droplet impingement at a velocity of 1 m/s onto a stationary wall.

Figure 9 shows that the heat flux tended to decrease by around 0.15 MW/m$^2$ over the duration of the simulation. The instantaneous heat flux was high initially, immediately after impingement, owing to the highly dynamic nature of droplet spreading and oscillations, as shown in Figure 10. The heat flux then gradually reduced with the loss of kinetic energy in the droplet. Similar heat transfer characteristics were reported by previous studies in the initial stages of spray impingement heat transfer and eventual evaporation. During the single-phase regime, until the droplet comes to rest, there is weak nucleation, and the heat transfer is dominated by impingement and single-phase film convection [22]. The time-averaged heat flux for the whole simulation was calculated to be 0.34 MW/m$^2$.

### 3.4. Non-Isothermal Impingement and Onset of Evaporation onto a Vibrating Boundary

Simulations again involving a droplet of 49 μm diameter were now of interest, involving phase change and a moving boundary. The focus remained on the heat transfer from the hot surface into the droplet and then into the surrounding vapor. Figures 11–13 show the surface-averaged heat flux (with a corresponding linear fit in each case) for boundary vibration amplitudes of 0.1 mm, 1 mm, and 10 mm, respectively, at frequencies of 10 Hz, 100 Hz, and 1000 Hz. Figures 14–16 show the corresponding dimensionless spreading diameters visualizing droplet patch oscillations.

Figure 11 shows that there was a similar trend in surface-averaged heat flux between the different frequencies for the same amplitude of 0.1 mm, i.e., the heat flux decreased over time. The lowest frequency at 10 Hz produced the largest amplitudes of surface-averaged heat flux, which was in line with the oscillations of a droplet after impingement, as shown in Figure 14. The oscillation amplitudes of surface-averaged heat flux reduced with increasing frequency (i.e., from 10 Hz to 100 Hz and to 1000 Hz). The linear fit to the data in each case showed a downward trend in surface-averaged heat flux which decreased with increasing frequency. For low-frequency vibrations, initial heat transfer was dominated by single-phase convection. When kinetic energy was lost, the droplet came almost to rest, which promoted the onset of evaporation. With increasing frequency, the droplet came to almost rest more quickly (although the droplet actually never came to rest; rather, the surface contact patch reached a quasi-static state even when there were oscillations within the volume of droplet).

The 1 mm amplitude results showed a similar pattern to the surface-averaged heat flux for the 0.1 mm amplitude case. The low-frequency wall vibrations resulted in large-amplitude oscillations of the surface-averaged heat flux. With increase in amplitude from 0.1 mm to 1 mm, there was a notable increase in the initial surface-averaged heat flux shortly after impingement, but the overall heat transfer was reduced owing to the difficulty that the droplets had in reaching a quasi-static state. This effect was not so evident at the highest frequency of 1000 Hz where the droplet reached a quasi-static state in a similar way to the 0.1 mm vibration amplitude case.

At a vibration amplitude of 10 mm, for frequencies of 10 Hz and 100 Hz, large-amplitude oscillations of the surface-averaged heat flux were seen immediately after impingement, resulting in difficulty for the droplet to reach a quasi-static state. This reduced the overall heat transfer. At the frequency of 1000 Hz, this effect was not evident because of high-frequency vibration appearing to quickly attenuate large-amplitude oscillations.

The overall time-averaged heat fluxes calculated for the stationary and moving boundary cases are given in Table 4 and shown graphically in Figure 17. At a boundary frequency of 10 Hz, Table 4 shows that the overall heat flux was lower than the stationary case for all amplitudes. The largest reduction in overall heat transfer could be seen to occur at an amplitude of 10 mm and frequency of 10 Hz, which was a reduction of 3.32% below the stationary case. The likely reason for this reduction is that large oscillations at low frequency delayed the droplet from reaching a quasi-static state. At the highest frequency of 1000 Hz, the overall heat flux was higher than the stationary case for all amplitudes (where the highest increase of 1.56% occurred at an amplitude of 1 mm). The likely reason for this is that the droplet washelped to reach a quasi-static state faster than in the stationary case.

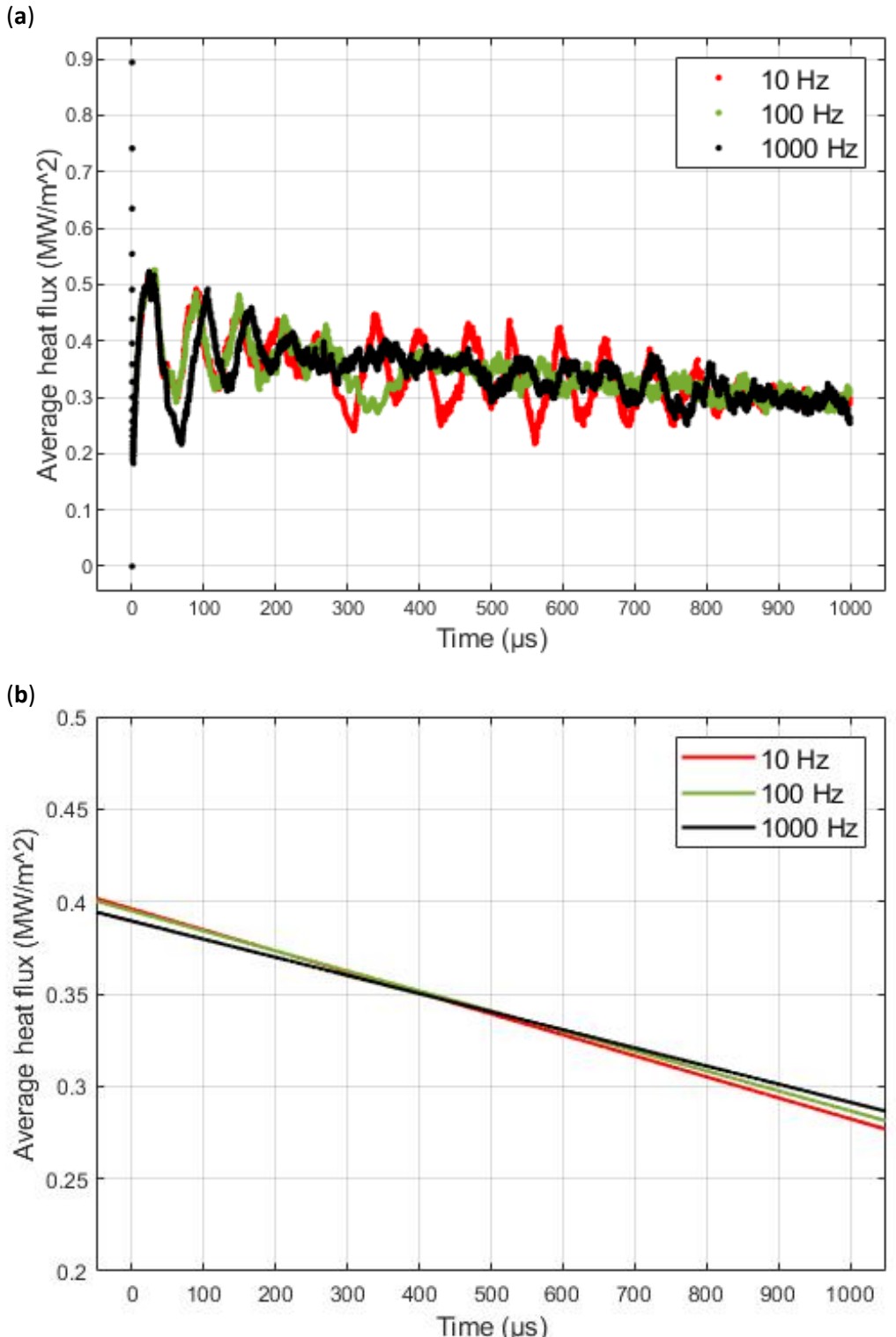

**Figure 11.** Surface-averaged heat flux for a surface vibration amplitude of 0.1 mm at frequencies of 10 Hz, 100 Hz, and 1000 Hz: (**a**) simulated values; (**b**) linear approximations to simulated values.

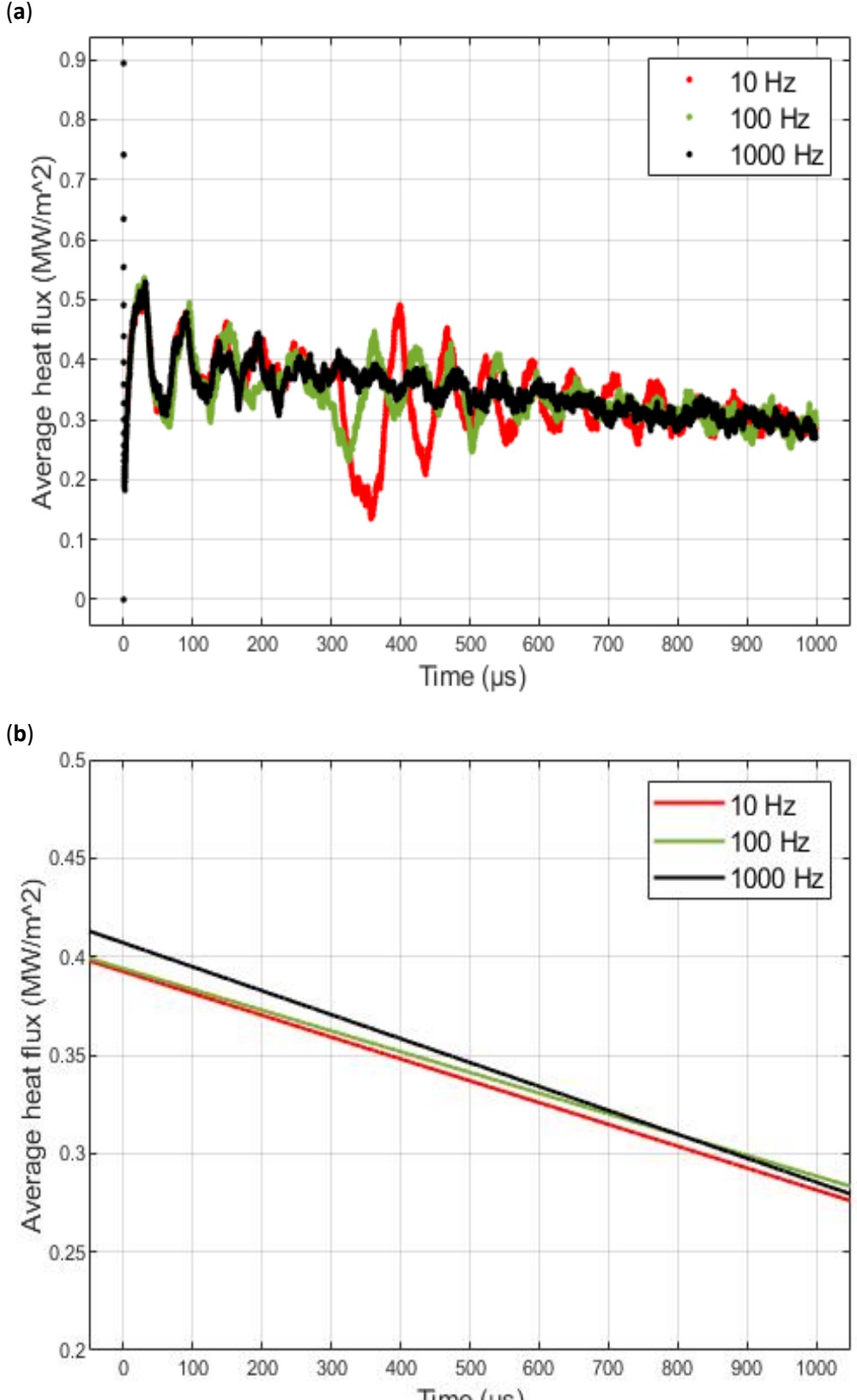

**Figure 12.** Surface-averaged heat flux for a surface vibration amplitude of 1 mm at frequencies of 10 Hz, 100 Hz, and 1000 Hz: (**a**) simulated values; (**b**) linear approximations to simulated values.

**(a)**

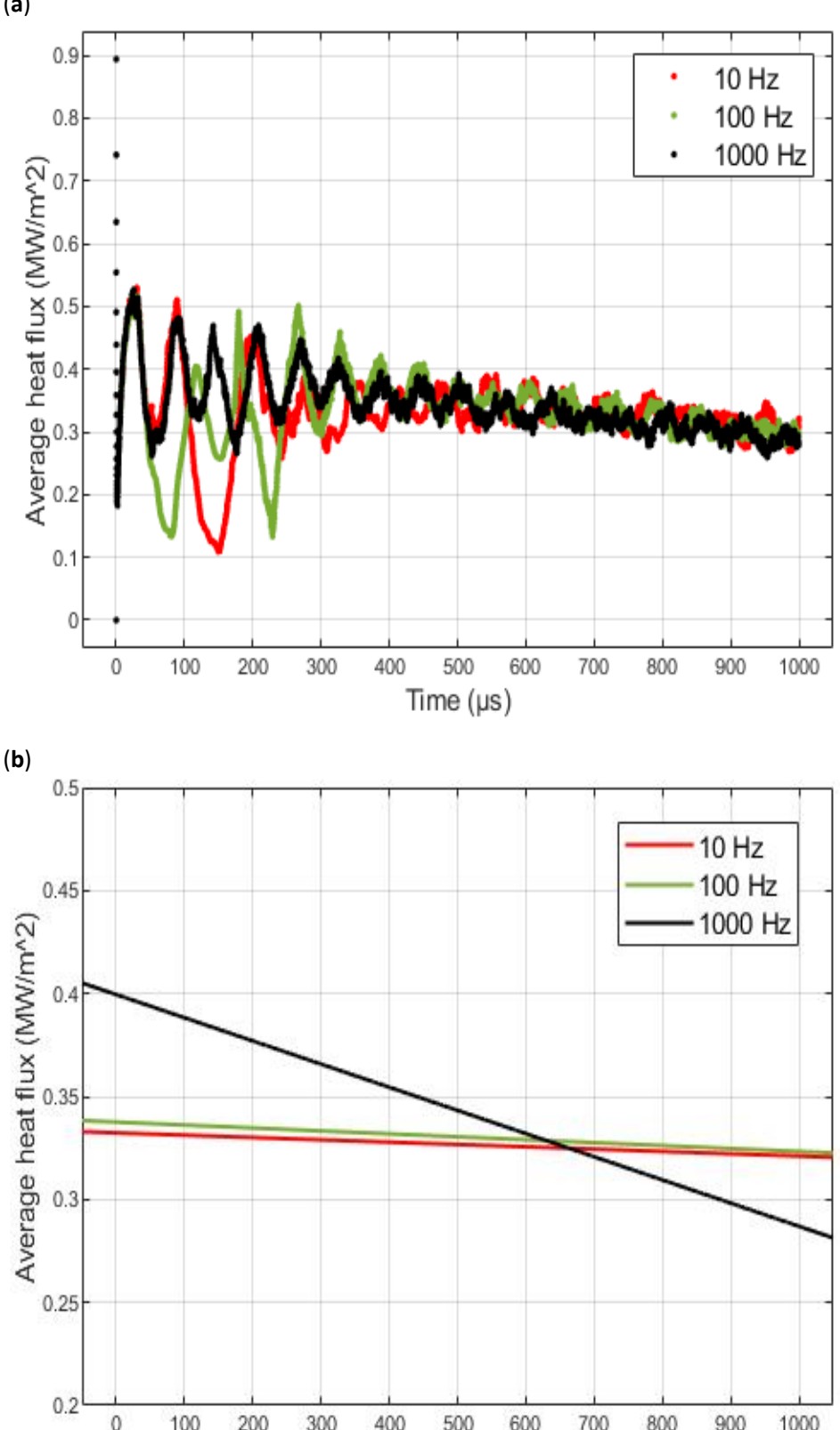

**(b)**

**Figure 13.** Surface-averaged heat flux for a surface vibration amplitude of 10 mm at frequencies of 10 Hz, 100 Hz, and 1000 Hz: (**a**) simulated values; (**b**) linear approximations to simulated values.

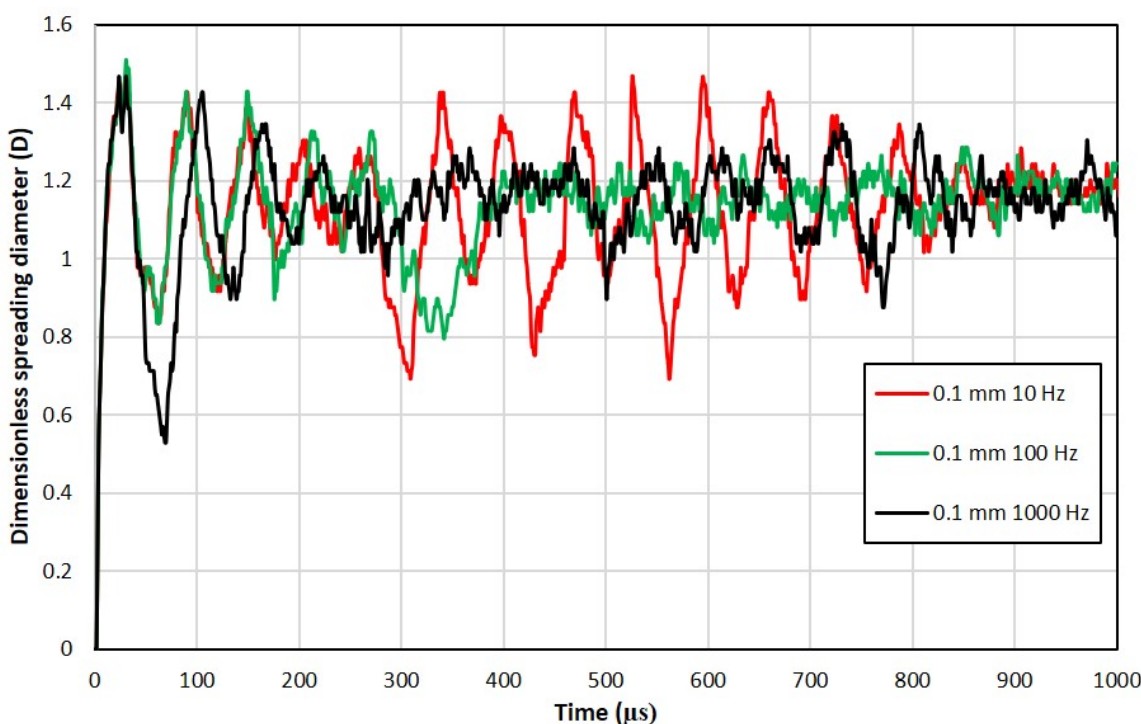

**Figure 14.** Droplet oscillations for a surface vibration amplitude of 0.1 mm at frequencies of 10 Hz, 100 Hz, and 1000 Hz.

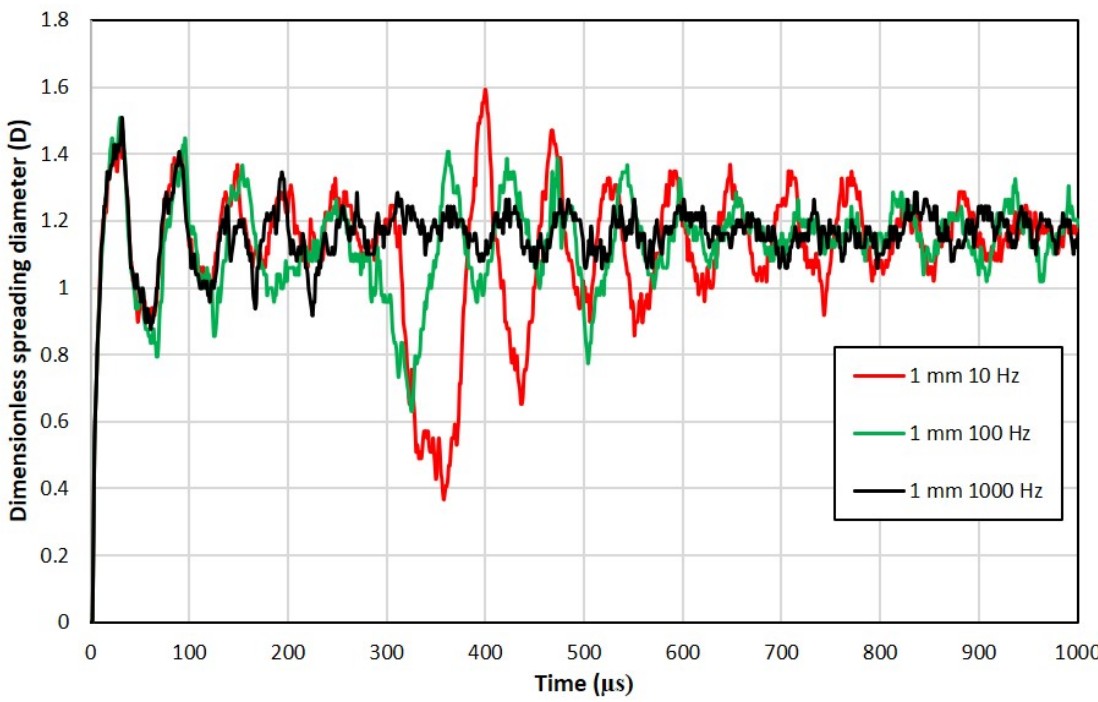

**Figure 15.** Droplet oscillation for a surface vibration amplitude of 1 mm at frequencies of 10 Hz, 100 Hz, and 1000 Hz.

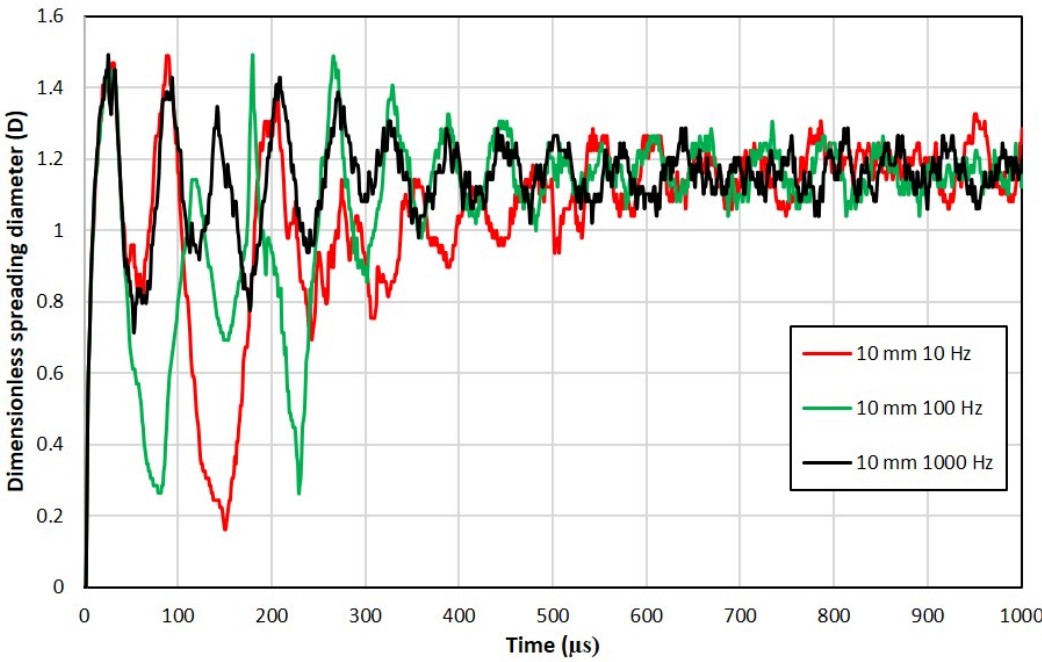

**Figure 16.** Droplet oscillation for a surface vibration amplitude of 10 mm at frequencies of 10 Hz, 100 Hz, and 1000 Hz.

**Table 4.** Time-averaged heat flux corresponding to stationary and dynamic simulations.

| Simulation No. | A (mm) | F (Hz) | $q_{ta}$ (MW/m$^2$) | % Difference in Heat Flux |
|---|---|---|---|---|
| C2 | 0 | 0 | 0.34 | 0 |
| D1 | 0.1 | 10 | 0.3383 | −0.5% |
| D2 | 0.1 | 100 | 0.3411 | +0.32% |
| D3 | 0.1 | 1000 | 0.3408 | +0.23% |
| D4 | 1 | 10 | 0.3349 | −1.5% |
| D5 | 1 | 100 | 0.3399 | −0.029% |
| D6 | 1 | 1000 | 0.3453 | +1.56% |
| D7 | 10 | 10 | 0.3287 | −3.32% |
| D8 | 10 | 100 | 0.3309 | −2.67% |
| D9 | 10 | 1000 | 0.3415 | +0.44% |

From the droplet oscillation shown in Figures 10 and 14, Figures 15 and 16, a general trend for droplet oscillation with respect to wall vibration frequency could be seen, which affected the overall heat transfer. For the lowest wall vibration frequency of 10 Hz, the droplets oscillated with large liquid–wall contact patch amplitudes at low frequencies for the three different wall vibration amplitudes. The large-amplitude liquid–wall contact patch oscillations at low frequencies resulted in reduced liquid–wall contact time, which in turn reduced the overall heat transfer. At the highest wall vibration frequency of 1000 Hz, the droplet oscillations had the highest frequencies with the lowest liquid–wall contact patch amplitudes for all the different wall vibration amplitudes. This resulted in the longest liquid–wall contact time and better overall heat transfer. The mid-wall vibration frequency of 100 Hz produced droplet oscillation characteristics in between the droplet oscillation characteristics generated by the lowest and highest wall vibration frequencies. Another important droplet oscillation characteristic observed was the difference in average liquid–wall contact patch areas for different frequencies. The highest frequency of 1000 Hz had the highest average liquid–wall contact patch area for all three wall vibration amplitudes. The larger average liquid–wall contact area also helped to achieve better overall heat transfer. The highest frequency of 1000 Hz also resulted in higher overall

heat transfer than the stationary case owing to the combination of larger contact area and longer contact time.

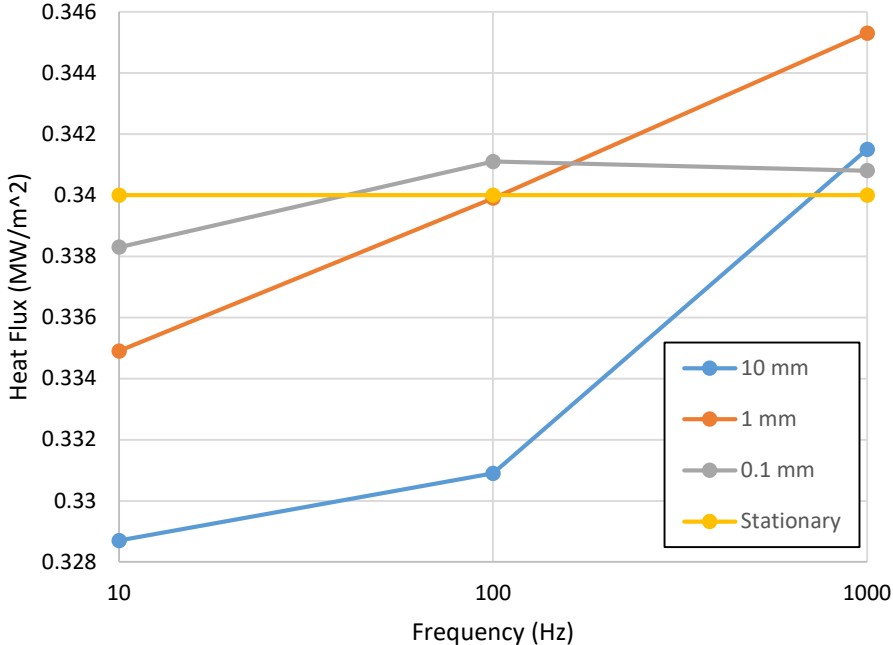

**Figure 17.** Time-averaged heat flux for single-droplet impingement onto stationary and moving boundaries.

## 4. Conclusions

A simulation capability was created to study the behavior of single-liquid-droplet impingement onto static and vibrating surfaces. The droplet fluid dynamic equations were solved using the Volume of Fluid method which includes the viscosity and surface tension effects of the droplet passing through surrounding air or steam, making contact with a possibly hot boundary. Initially, the dynamic behavior of isothermal impingement was examined, first without and then with boundary vibration for a droplet moving through air. Simulations were then undertaken to assess heat transfer with the liquid droplet initially at saturation temperature passing through steam and making contact with a hot vibrating boundary in which droplet evaporation commenced.

The isothermal simulations were used to establish how both rebound conditions and the time interval between initial contact to detachment varied with droplet diameter for droplet impingement onto a stationary boundary. It was found that the minimum impact velocity for rebound reduced linearly with droplet diameter. By contrast, it was found that the time interval between initial contact and detachment appeared to increase linearly with droplet diameter. Isothermal simulations involving a vibrating surface showed that the relative minimum impact velocity for rebound (i.e., the droplet velocity plus the boundary velocity) was higher than the minimum impact velocity for static boundary droplet rebound. Droplet impingement simulations which included the onset of evaporation showed that large-amplitude surface vibration appeared to reduce heat transfer, whereas low-amplitude high-frequency vibration seemed to increase heat transfer.

**Author Contributions:** Conceptualization, J.F.D.; formal analysis, J.T.J.; investigation, J.T.J.; methodology, J.F.D.; validation, J.T.J.; writing—original draft, J.T.J.; writing—review and editing, J.F.D. All authors read and agreed to the published version of the manuscript.

**Funding:** This research received no external funding.

**Conflicts of Interest:** The authors declare no conflict of interest.

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
