# Peer review of "Numerical Simulation of Single-Droplet Dynamics, Vaporization, and Heat Transfer from Impingement onto Static and Vibrating Surfaces"

_fluids, doi:10.3390/fluids5040188_

Round 1

Reviewer 1 Report

In this paper, the dynamics and evaporation of a droplet impinging onto stationary and vibrating boundaries are simulated using a 2D axisymmetric model and the volume-of-fluid (VOF) method. This paper contains some very interesting results, however, to make this manuscript suitable for publication the authors need to address the following comments:

  1. Line 111: please clarify the ANSYS-FLUENT version in the manuscript
  2. Line 119: please check the equation; the units on the left and right sides are not identical.
  3. Lines 139-140: what is the correlation between kl (the surface curvature) and R1, R2?
  4. Lines 159-162: It seems that Ce and Cc (the constants) have a unit like 1/s. Please check it. Otherwise, the units on the left and right sides of equation (7) (line 155) would not be identical.
  5. Line 194: it is claimed that “Geo-Reconstruct is the most accurate interpolation method…”. Is there any study about comparing different methods (different algebraic VOFs, CLSVOF, etc.) and finding the most accurate one!!! If so, please cite that and explain that study in details. If not, just remove this sentence.
  6. Section 2.4: In my opinion, more information about Blake and Coninck dynamic contact angle model is necessary. For example, I couldn’t understand how is the dynamic contact angle in equation (10) calculated? Why are equations (11) and (12) implemented in the UDF? I think, equation (10) should be used to calculate the contact line velocity. In general, this part is a bit confusing and the authors should prepare a schematic figure to show how their algorithm works. More information about the UDFs is also necessary.
  7. Lines 298-301: Add citations for this sentence.
  8. Line 307: is it static contact angle?
  9. Figures 5 and 6: for these figures, calculate and report Reynolds and Weber numbers. Add a short discussion about critical Weber and Re.
  10. Section 3.2: the droplet shape at different A, F and V(wall) as a function of time should be presented (like figure 5).
  11. Section 3.3: what are the values of thermophysical properties (such as densities, viscosities, surface tension, thermal conductivities) here.
  12. Section 3.4, page 18: present a figure for the droplet shapes (like figure 5). In this case, discussing and understanding the physical phenomena would be much easier.

Reviewer 2 Report

The authors presented a well-defined simulation but the results are not analyzed well enough from physics point of view. I suggest that major revision is needed to meet the standard of the journal. Following questions also need to be answered in the revised paper.

  1. The values of Ce and Cc are both set as 0.1, but the actual coefficients may vary, the influences of these factors should be studied.
  2. The first figures of Fig. 5 are not the instant of first contact, but the description presented the time interval between first contact and detachment, which may lead to misunderstanding.
  3. What are the velocities of droplets in Fig. 7? If the velocities are not identical, what is the meaning of this comparison and linear fit? How will the detachment time change if the impact velocities are increased?
  4. Why did rebound happen in B10 but not in B4 where the impact velocity was higher?
  5. Why was the droplet velocity for non-isothermal simulations reduced to 1 m/s?
  6. Why was the high initial heat flux owing to the highly dynamic nature? Is there any specific analysis or quantitative theory?
  7. How does the relationship between droplet self-oscillation frequency and wall oscillation frequency influence the overall heat transfer and droplet dynamics? This needs to be discussed in detail.
  8. Why is gravity effect not included?
  9. In Table 2: Case B1~B3 have no difference with B4~B6, so are B7~B9/B10~B12. In Table 3 it is corrected.
  10. Line 83: … is (a) widely … fluids( )where…
  11. Line 212, 257, 264, 279, 320, 329, 334, 335, 341, 349, 375, 380, 389, 390, 405: reference error.
  12. Font sizes are not consistent.

Round 2

Reviewer 1 Report

The manuscript in its current format is suitable for publication. However, in Eq. 6, there is a typo error.